# Precise Design of Alginate Hydrogels Crosslinked with Microgels for Diabetic Wound Healing

**DOI:** 10.3390/biom12111582

**Published:** 2022-10-28

**Authors:** Yishu Yan, Panpan Ren, Qingqing Wu, Tianmeng Zhang

**Affiliations:** 1School of Life Science and Health Engineering, Jiangnan University, No. 1800, Lihu Avenue, Wuxi 214122, China; 2Bloomage Biotechnology Corporation Limited, Jinan 250101, China

**Keywords:** alginate hydrogel, tissue regeneration, microgel, scaffold, diabetic wound healing

## Abstract

Alginate hydrogel has received great attention in diabetic wound healing. However, the limited tunability of the ionic crosslinking method prevents the delicate management of physical properties in response to diverse wound conditions. We addressed this issue by using a microgel particle (fabricated by zinc ions and coordinated through the complex of carboxymethyl chitosan and aldehyde hyaluronic acid) as a novel crosslinker. Then the cation was introduced as a second crosslinker to create a double crosslinked network. The method leads to the precise regulation of the hydrogel characters, including the biodegradation rate and the controlled release rate of the drug. As a result, the optimized hydrogels facilitated the live-cell infiltration in vitro and boosted the tissue regeneration of diabetic wounds in vivo. The results indicated that the addition of the microgel as a new crosslinker created flexibility during the construction of the alginate hydrogel, adapting for diverse applications during diabetic-induced wound therapy.

## 1. Introduction

Diabetic wounds are a serious complication, associated with a high rate of infection, amputation or losses of limbs, and resulting in high health care burden and poor quality of patient life [1]. Hydrogel is considered as an ideal scaffold for diabetic wound care due to its similarity to the extra cellular matrix (ECM) [2]. In particular, alginate hydrogels have drawn great attention because of their high biocompatibility [3]. The hydrophilic nature of the wound dressings creates the moist wound environment as required by diabetic wound healing. Moreover, they easily crosslink with other materials that promote wound healing in clinical applications [3]. So far, alginate hydrogels have been one of the most popular biomaterials for diabetic wound management.

Regardless of the prevalence, the current alginate hydrogels are only able to offer some success with regard to different cases. For one thing, wound healing undergoes exudative, inflammatory, proliferative and regenerative stages sequentially, and a variety of cellular and matrix components are involved in it. For another, each ECM is distinct across pathological tissues [4]. The situation of a diabetic wound is more complicated due to the presence of an ulcer, osteomyelitis or gangrene [5]. Therefore, the desired scaffolds should be precisely tuned in terms of their physical characters according to the specific conditions and stages of the wounds.

However, the current alginate hydrogels are formed by adding divalent cations as crosslinkers. The ionically crosslinked networks have shown limited adjustability. For example, the current alginate hydrogels exhibit a very slow degradation rate in the physiological environment because of the uncontrollable removal of the cations. The processes usually last over two months after gel implantation, inhibiting cell infiltration and remodeling in diabetic wounds [5,6]. Additionally, the cation crosslinking methods also display limited adaptable features including the mechanical properties, swelling behaviors and controlled release of the drugs encapsulated in the biomaterial.

Microgels are regarded as soft colloids due to their strong ability of deforming, swelling/deswelling and interpenetrating with other microgels [7]. They have served as crosslinkers to accurately control the gel’s physical characters for various cellular microenvironment recaptures [8,9,10]. Therefore, we hypothesize that the alginate hydrogel crosslinked with the flexible microgels would possess improved tunability, thus making it adaptable for diverse wound conditions.

As a proof of the concept, we designed a series of alginate hydrogels using a biocompatible microgel as a crosslinker, which was synthesized by coordinating Zn^2+^ to the mixture of carboxymethyl chitosan (CMCS) and aldehyde hyaluronic acid (A-HA) [11]. Then a double crosslinked network was set up by adding the divalent cation as the second crosslinker. The microgel combined with the cations provided multi-scale regulation methods of the hydrogels, resulting in a precisely controlled degradation rate, mechanical strength and a controlled drug-release rate. As a result, the optimized hydrogel enabled the live-cell accommodations and boosted the tissue regeneration of diabetic wounds in vivo, indicating a high potential for application.

## 2. Materials

### 2.1. Materials

Sodium alginate was purchased from Aladdin Industrial Co., Ltd. (Shanghai, China); chitosan was purchased from National Pharmaceutical Group Chemical Reagent Co., Ltd. (Shanghai, China); hyaluronic acid was a gift from Bloomage Biotech. Co., Ltd. (Jinan, China). The monochloroacetic acid (MCA), isopropyl alcohol, and ZnCl_2_ were purchased from National Pharmaceutical Group Chemical Reagent Co., Ltd. (Shanghai, China); the fetal bovine serum (FBS) and Dulbecco’s modified essential medium (DMEM) culture were purchased from Thermo Fisher Scientific Inc. (Waltham, MA, USA). FITC was purchased from Xi’an Ruixi Bio. Tec. Co., Ltd. (Xi’an, China). Dihydrochloride (DAPI) and lysosome tracker were purchased from Beyotime Bio. Tec. Co., Ltd. (Nanjing, China). *Streptozotocin* (STZ) was purchased from Alladin Tec. Ltd. (Shanghai, China). All the other reagents, solvents and chemicals were of analytical grade.

### 2.2. The Cells and Animals

The mouse epidermal fibroblast cell line (NIH-3T3) was obtained from China Center for Type Culture Collection. The cells were cultured in DMEM supplemented with 10% FBS, 4 mM L-glutamine, streptomycin and penicillin, in an incubator with humidified atmosphere containing 5% CO_2_ at 37 °C.

Male C57B6/J mice (6 w, around 20 g) were purchased from Lingchang Biotech Co., Ltd. (Shanghai, China). All the animal experiments were performed in accordance with the principles of care and use of laboratory animals and were approved by the ethics committee of Jiangnan University (JN. No20201215b0600218[358]).

## 3. Methods

### 3.1. The Hydrogel Preparation and Characterization

#### 3.1.1. General Methods

The ^1^H NMR analysis was carried out on a 400 MHz Bruker NMR spectrometer (Appendix A) [12]. The aldehyde group content of A-HA was determined with 3,5-dinitrosalicylic acid assay method [13]. The substitute degree was determined as 23.4%. The FT-IR spectra were performed on an iS5 FT-IR instrument (Thermo Fisher, Waltham, MA, USA) and scanned from 400 to 4000 cm^−1^.

The relative molecular weight and distributions of the polysaccharides were determined on a Waters 1525 high performance liquid chromatograph equipped with the column of Ultrahydrogel™ Linear (300 mm × 7.8 mm) (Appendix A). The column temperature was kept at 40 °C. Sodium nitrate solution (0.1 M) was used as the mobile phase with the flow rate of 0.8 mL/min. The standards (dextran T-2000, Dextran T-300, Dextran T-150, Dextran T-10 and Dextran T-5) were purchased from the National Institute for the Control of Pharmaceutical and Biological Products, with the Mw of 2,000,000, 30,060, 135,030, 9750 and 2700, respectively.

#### 3.1.2. ^1^H NMR Analysis for the G/M Ratio of Sodium Alginate

The G/M ratio of the sodium alginate was analyzed according to the published paper [14]. The pH of the alginate solution (1%, 10 mL) was adjusted to 5.6 using hydrochloric acid (0.1 M) and heated in boiling water for 1 h. The resulting solution was adjusted to pH 3.8, heated in boiling water for another 30 min, neutralized to pH 7 (with sodium hydroxide, 0.1 M) and freeze dried.

The sample was re-dissolved in D_2_O (5 mL), and 0.7 mL of the solution was further mixed with TTHA (triethylenetetraminehexaacetic acid, 0.15 M, 40 μL) solution. The pH was adjusted to 5.5 using NaOD (0.1 M) solution. ^1^H NMR spectra were recorded at 80 °C. The anomeric proton signals were compared with the standard spectra (Appendix A).

#### 3.1.3. The Microgel Preparation

CMCS was synthesized according to the published method [11]. The carboxymethyl substitute degree was determined as 58%. Hyaluronic acid (1 g, 1 L) was reacted with NaIO_4_ (500 mg) for 2 h, dialyzed against water (3500 Dw), and lyophilized to obtain A-HA [15].

The microgels were prepared as followed (Figure 1). The powder of A-HA (50 mg) was incubated with CMCS (200 mg) in 5 mL DI water for 2 h, followed by the addition of ZnCl_2_ (300 mg, 0.5 mL). The reaction solution turned opaque immediately, and the color changed to light yellow. The mixture was stirred overnight. Then, the solution was centrifuged for 5 min at 6000 rpm/min. The gel-like sediment was harvested and washed with DI water for 3 times. The obtained microgels (Zn^2+^-CMCS-A-HA) were dispersed in water (5 mL) for further treatment.

The microgel solution were diluted with pure water to a concentration of 100 ng/mL. The dynamic diameter was determined by Dynamic Light Scattering (DLS) with a Zetasizer Nano ZS system (Malvern, Herrenberg, Germany) equipped with a 633 nm He-Ne laser. The microgels containing water were dipped on a carbon-coated copper grid. Then the samples were air dried and observed with a transmission electron microscope (TEM, Japanase Electronic, JEM-2100F).

#### 3.1.4. The Alginate Gel Fabrication

The microgels were crosslinked with alginate through the electrostatic interaction between the CMCS on the surface of the microgels and the alginate, mediated by D-glucono-δ-lactone (GDL) (Figure 2) [16,17]. The GDL hydrolyzed and released protons, which protonated the amino groups of the CMCS. Then, the protonated CMCS and the alginate would crosslink to form hydrogels through electrostatic interaction.

Briefly, alginate sodium (1%, 2 mL) was mixed with different weights of fresh prepared microgels (0.5 mg, 1 mg, and 2 mg). Fresh saturated GDL (15 μL) solution was added. The pre-gel solutions were stored at room temperature for 10 min until the hydrogels formed, which were named as A, B, and C, respectively. To build the double crosslinked network, the hydrogel of A was immersed in a 1% CaCl_2_ (10 mL) buffer overnight. The aqueous solution was removed, and the hydrogel was named as D. The alginate sodium hydrogel (named as AS) crosslinked with only the cations was prepared by adding the alginate sodium solution to the CaCl_2_ (1%) solution. The gels were lyophilized, and their microstructures were observed by Scanning Electronic Microscope (SEM, Hitachi S-2400, Tokyo, Japan).

### 3.2. In Vitro Swelling Assay

The lyophilized hydrogels (50 mg) were immersed in distilled water (50 mL) at room temperature for 1 day. The gels were removed from the water at a fixed time and weighed. The swelling ratio (%) = (Wt − W_0_)/W_0_ × 100. W_0_ and Wt represented the original weight of the microgels and the weight at time t, respectively. The experiments were repeated three times.

### 3.3. In Vitro Drug-Release Assay

BSA was selected as the model molecule to test the drug-release behavior in the hydrogels. As a protein, the amino group on the FITC-BSA could reversely react with the aldehyde group on A-HA through the formation of the Schiff base bond, thereby easily incorporated in the microgels. A-HA (5 mg, 5 mL) was incubated with the FITC-BSA (0.05 mg) for 20 min before the hydrogel A, B, C and D preparation. The AS gel incorporating the FITC-BSA was also prepared for comparison. These hydrogels (10 mg) were immersed in the release buffer (HEPEs, pH 7.4, 10 mL) while gently stirring at room temperature. The supernatant (0.5 mL) was withdrawn at different time intervals. The relative fluorescence intensity (with the HEPEs buffer as control) of the FITC-BSA was determined with photoluminescence (Hitachi, F-7000; Excitation, 490 nm; Emission, 520 nm). The amount of released BSA was calculated according to the standard curves. The experiments were repeated three times.

### 3.4. Rheological Property Determination

The rheological properties of the hydrogels (A, B, C and D) were determined using a rheometer (DHR, TA instrument, Newark, NJ, USA) on the 40 mm diameter parallel plate, with a default gap of 0.5 mm. Frequency-sweep measurements were carried out from 0.5 to 100 rad/s in the linear viscoelastic region via dynamic strain sweep measurements at room temperature. The experiments were repeated three times.

### 3.5. In Vitro and In Vivo Degradation Assay

The fresh prepared hydrogels (A, C, D) were weighed (W_0_), and incubated with an HEPE (pH = 7.0) buffer at 37 °C in a shaking table (200 rpm/min). The gels were taken out to weigh (the weight was determined as Wt) at predetermined times during a 10-day period. The cumulative degradation degree in vitro was calculated as (1 − Wt/W_0_) × 100%.

The hydrogels were cut into the small cylinder with the diameter of 0.9 cm and the height of 0.3 cm. The C57B6/J mice were anesthetized with Avertin (240 mg/kg). Incisions were made on the back of the mice. The cylinders were implanted subcutaneously. The skin of the mice was sewed up. Then the mice were kept warm on a pad for a while until recovery and returned to normal feeding conditions. The mice were sacrificed at Day 20 after surgery. The tissue of the incision was taken out and fixed in 4% paraformaldehyde solution for H&E staining. The slices were analyzed on a light microscope.

### 3.6. 3D Cell Culture in Hydrogels

The cells were pre-mixed with the alginate solution at a density of 200,000 cell/mL. Then different concentrations of the microgels were added, followed by the addition of the GDL solution. The mixture was placed in the incubator, and the cell-loaded hydrogel of A, B and C formed within 10 min. The cells were cultured for 72 h. The live cells were stained with Lysotracker red and DAPI. The 3D cell cultures were photographed with a laser-scanning confocal microscope (TCS SP8, Leica, Germany).

### 3.7. The Whole Cortex Defect Model on Streptozotocin (STZ) Induced Diabetic Mice

The animal model was created according to the published methods with some modifications. Briefly, male C57B6/J mice (6 weeks) were intraperitoneally injected with STZ (70 mg/kg) for four consecutive days and fed a normal diet for another ten days. The mice with a blood glucose level over 13.5 mmol/L were chosen for preparing the whole cortex defect model. The selected mice were anesthetized with Avertin (240 mg/kg), and excisional full-thickness skin removal wound splinting (0.9 cm × 0.9 cm) was performed in the middle of the back [18].

The treated mice were randomly divided into four groups (9 mice for each group). The group with the wound fixed with medical proof alone was regarded as the negative control (NC). The other groups were covered with the hydrogels of C, D or AS on the wounds, and double fixed with medical proof fabric. The size of the wounds was determined, with photographs were taken on alternative days during the wound repair.

To track the epidermis re-epitelization process during the wound healing, some of the mice (*n* = 3) were killed on Day 3, and others were killed on Day 5 (*n* = 6). The wound tissues were removed and fixed in a 4% paraformaldehyde buffer. Then H&E staining and Masson’s trichrome analysis were performed to evaluate the wound-healing rate.

### 3.8. Statistical Analysis

Data were expressed as mean ± SEM. Difference among groups was assessed using one-way variance followed by Bonferroni’s Multiple Comparison Test. Results were considered statistically significant at *p* < 0.05.

## 4. Results and Discussions

### 4.1. The Hydrogel Preparation and Characterization

The microgels were synthesized before the hydrogel preparation. The diameter was found to be in the range from 1–5 μm by the DLS spectrophotometer (Appendix A). The TEM photographs (Appendix A) showed the microgels were highly uniform with the diameter no more than 100 nm (the size highly shrunk after the water was evaporated).

Immediately after the microgels were prepared, the hydrogels (A, B and C) were fabricated by incorporating the microgels into the bulk gel. To further prove that the hydrogels allow CaCl_2_ as the second crosslinker, the hydrogel of A was selected to create the double crosslinked network (D)_._ The FT-IR spectra of the hydrogels were recorded and compared (Figure 3). The main peaks did not show significant variations. Stretching vibrations of O–H bonds were observed at 3000–3600 cm^−1^. Stretching vibrations of aliphatic C–H were observed at 2920–2850 cm^−1^. The peaks around 1650 and 1450 cm^−1^ were attributed to asymmetric and symmetric stretching vibrations of carboxylate salt ion, which showed little red shift in the spectra of hydrogel D compared with A, B and C, indicating the formation of the cation crosslinked alginate network [19].

The morphology of the hydrogels was investigated with SEM (Figure 4a–c). The hydrogels of A and B showed similar honeycomb-like structures. The hydrogel of C showed the intermediate state of the honeycomb and laminated structures, whereas D showed a typical laminated morphology (Figure 4d). Additionally, all the microgels were embedded in the hydrogel matrix. The crosslinking density increased significantly as the microgel concentration increased, with the average pore size decreasing gradually. The diameter of the pore size was 25 μm for A, 15 μm for B, and no more than 10 μm for C (Appendix A).

The swelling properties of hydrogels play a critical role in absorbing extrude in the diabetic wound [20]. In this study, all the hydrogels displayed similar swelling kinetics (Figure 4e). The final swelling percentage is about 550% for A, increased to 600% for B and C, and 700% for D. The final swelling ratio did not reduce as expected when the crosslinking degree increased. We supposed the microgels composed of highly hydrophilic polysaccharides (CMCS and A-HA) collaboratively contributed to the swelling behavior of the alginate hydrogels. In addition, the hydrogel D had a significantly higher swelling ratio than the hydrogels A, B and C. The reason could be attributed to the fact that the electrostatic interactions, which are the main driving force for the hydrogel formation, would be disrupted during lyophilization, and would not be completely reconstructed after the lyophilized product came into contact with water.

The rheological assay was performed (Figure 4f) 30 min after the GDL addition. The results showed that both the storage modulus (G′) and loss modulus (G″) were independent of the frequency, indicating that the hydrogels were structurally stable. All the samples showed viscoelastic gel properties (G′ >> G″) over the tested frequency range.

In addition, the storage modulus dictating the mechanical character of the biomaterial is the key property affecting tissue regeneration efficiency. The value of A was 100 and increased to 700 for C, whereas the value of D increased to over 1000. The results showed that the mechanical strength was enhanced as the crosslinking density increased. Moreover, the double network further reinforced the mechanical strength. Thus, the combinational effects of the microgels and the cations enabled a highly tunable mechanical strength, meeting the diverse requirements of diabetic wound therapies.

### 4.2. The Drug-Release Determination

Strategies to precisely control the drug (such as b-FEF) release kinetics over an extended time remain a key issue for diabetic wound healing [21]. The porous structure and swelling property enabled hydrogels as important drug-sustained released platforms [22]. Nonetheless, these conventional structures might lose the ability to control the drug-release rate because the low crosslinking density leads to a high percentage of drug leakage before arriving at the disease site [23]. As for the microgel-in-gel structures, the microgel could be used as the first impounding reservoir of the drugs before they diffused into the network of the bulk gel, thereby providing a more meticulous drug-release-rate control [24].

We used FITC-BSA as a model drug to investigate the drug-release behavior (Figure 5). Indeed, the initial FITC-BSA release rate of A, B, C, and D was far below that of AS, avoiding the “burst release” effect induced by the sudden rise of the drug concentration. Secondly, the drug-release rate was reversely related with the microgel concentration (A > B > C) due to more FITC-BSA entrapment in the microgels as their concentration increased.

By contrast, the drug-release rate of the hydrogel D increased substantially. Because the cations have strong interaction with the CMCS, the structure of the microgels in the bulk gel would be destroyed after being double crosslinked, losing the function as the drug impounding reservoir, and hence the drug-sustained-release capability. Indeed, although the drug-release rate decreased at first, the accumulating released drug concentration of D was equal to that of AS in a 120-h period. Taken together, the results indicated that the microgels could entrap the drug in the networks, and precisely control the drug-release rate compared with the hydrogel that was crosslinked with the cations alone.

### 4.3. The Hydrogel Degradation In Vitro and In Vivo

Controlled biodegradation of hydrogels is a highly desirable property for diabetic wound healing. A different degradation rate is required to control the diverse cell microenvironment for customized wound therapy [25]. To investigate the degradation behavior in vitro, the hydrogels were incubated in the HEPE buffer with violet shaking and subjected to weight measurement at a fixed time (pH = 7.0). As shown in Figure 6a,b, the hydrogels of A and B completely fell apart by mechanical force within 4 days due to weak mechanical strength, and C also degraded during a 9-day period. By contrast, D and AS degraded in the first 4 days but did not continue their weight loss in the following period.

The hydrogels were then implanted in the subcutaneous tissues of mice. After 21 days, the tissues were taken out and H&E staining was performed. As shown in Figure 6c, the gel of A already completed the biodegradation process and was replaced with the native regenerated tissue. The gel of C was under the degradation process. By contrast, D had not been degraded yet. The results strongly confirmed the microgel as a crosslinker-endued alginate hydrogel with a tunable biodegradation capability. The degradation time prolonged as the concentration of microgels increased.

Moreover, the cell encapsulation property is indicative of the ability for promoting tissue regeneration [26]. As shown in Figure 7, the surrounding cells of the regeneration tissues directly infiltrated the inside of the undegraded material in the hydrogel milieu of C, indicating that the hydrogel provides physical support for cell growth. However, the cells accumulated only alongside the edge of the material D without the inside area, in coincidence with the results that the small pore size of hydrogel D would not allow the growth of cell.

### 4.4. The 3D Cell Cultures in Hydrogels

To further demonstrate that the hydrogels enabled the cell accommodation, NIH-3T3 cell lines were seeded on the microgels before the hydrogel preparation. The cells containing the hydrogels were observed over a 72-h period with DAPI and Lysotracker staining. As shown in Figure 7a, the cells (stained with DAPI) were widely spread inside the hydrogels. The hydrogel A incorporated a higher density of cells (two times, Figure 7b) compared with B and C, due to the low steric hindrance induced by the larger pore size. Moreover, the lysosome activity was affected by the microgel concentration, because the percentage of the cells stained with both DAPI and Lysotracker decreased as the microgel concentration increased.

These results re-confirmed the hydrogels could work as a scaffold for 3D cell culture [27]. In diabetes, normal progression through the normal wound-healing phases is impaired, resulting in dysfunctional tissue regeneration, also known as epithelialization processes [28]. The ability of the expedient networks to enable cell accommodation probably led to the epithelialization acceleration in diabetic wounds.

### 4.5. The Hydrogels Promoted Tissue Regeneration of the Diabetic Wound In Vivo

To test our hypothesis, we established a full excision wound on STZ-induced diabetic mice, implanted the hydrogels crosslinked only with microgels and the double crosslinked network (D) on the wound, and observed the wound-healing process during a 5-day period (Figure 8a,b). Balancing the mechanical strength as well as the cell encapsulation ability, the hydrogel of C was selected to represent the microgel crosslinked hydrogel. Both C and D accelerated the wound closure compared with the NC group.

However, the H&E staining (Figure 8c) confirmed that much more regenerate tissue deposited on the wound bed of the C group at Day 5. Moreover, Masson’s trichrome (Figure 8d) staining demonstrated that collagen, a main component of the re-epithelialized tissue, had been synthesized and excreted, whereas the D and NC group had not begun the tissue-repair process yet. Therefore, we confirmed that C accelerated the epithelialization of diabetic wound healing, consistent with previous studies that C allowed for better cell infiltration. Although D also showed an increased wound-closure rate, the biomaterial only provided a moist environment for absorbing the wound extrude. Therefore, the two hydrogels accelerated wound closure through completely different mechanisms.

## 5. Conclusions

In this paper, we provided a new class of alginate hydrogels using the house-made microgel (Zn^2+^-CMCS-A-HA) as a crosslinker. This microgel-in-gel design allowed for facile approaches for the micro hierarchical structure control by modulating the concentration of both the microgel and the cation simultaneously. As a result, the hydrogels exhibited highly tunable mechanical properties, obviously accelerating the biodegradation rate and the precise release rate of the drugs encapsulated in the biomaterial.

Meanwhile, the gels served as physical support for 3D cell culture in vitro. In vivo, the cells were able to infiltrate the scaffold along with the hydrogel degradation. Ultimately, the optimized hydrogel led to early epithelialization and wound closure on a diabetic wound compared to the cation crosslinked hydrogel. The results indicated that the hydrogels were promising in diabetic wound healing application.

In fact, the microgels allowed for more chemical modulations and resulted in much higher tunability. By using the distinct microparticles as crosslinkers, the alginate hydrogels would display various chemical and physical properties, enabling much more adaptable characteristics for diabetic wound-healing processes.

## Figures and Tables

**Figure 1 biomolecules-12-01582-f001:**
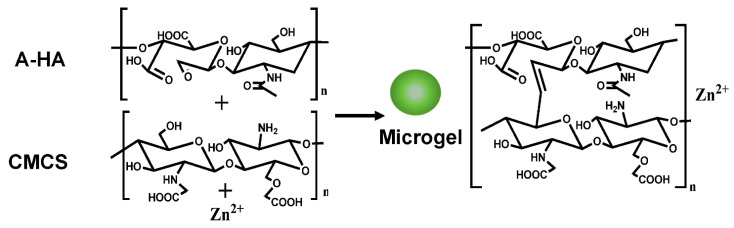
The scheme of the microgel synthesis.

**Figure 2 biomolecules-12-01582-f002:**
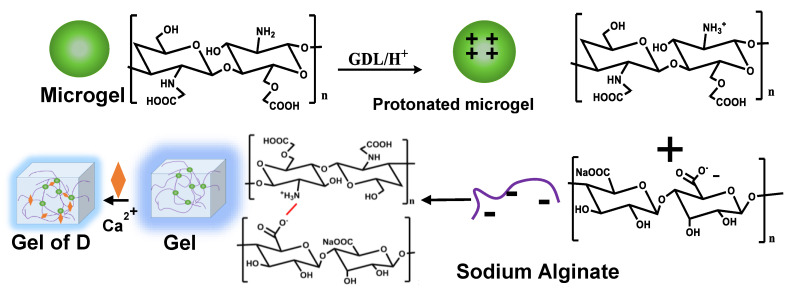
The scheme of hydrogel preparation.

**Figure 3 biomolecules-12-01582-f003:**
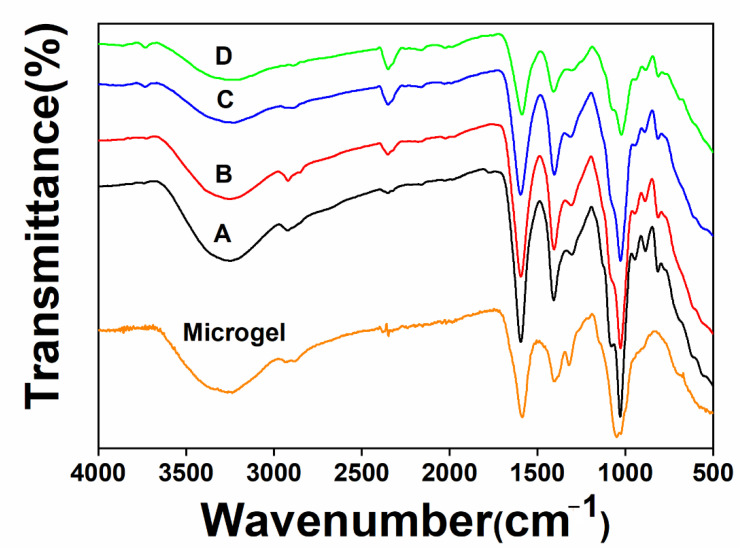
FT-IR spectra of the microgels and hydrogels.

**Figure 4 biomolecules-12-01582-f004:**
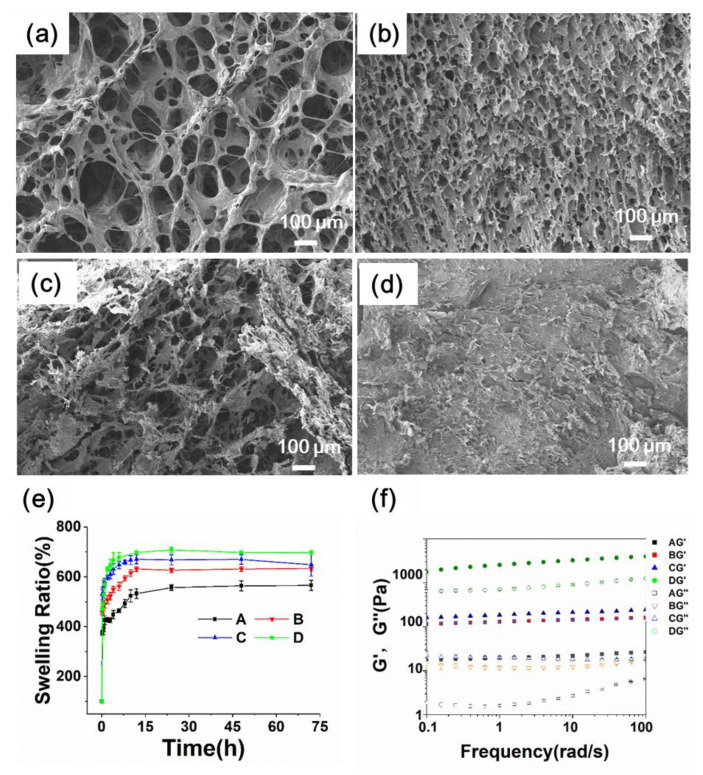
Characterization of the hydrogels. (**a**–**d**) SEM images of the hydrogels of A (**a**), B (**b**), C (**c**), and D (**d**). (**e**) The swelling behavior of the hydrogels. (**f**) Frequency-sweep dynamic rheological profiles of the hydrogels.

**Figure 5 biomolecules-12-01582-f005:**
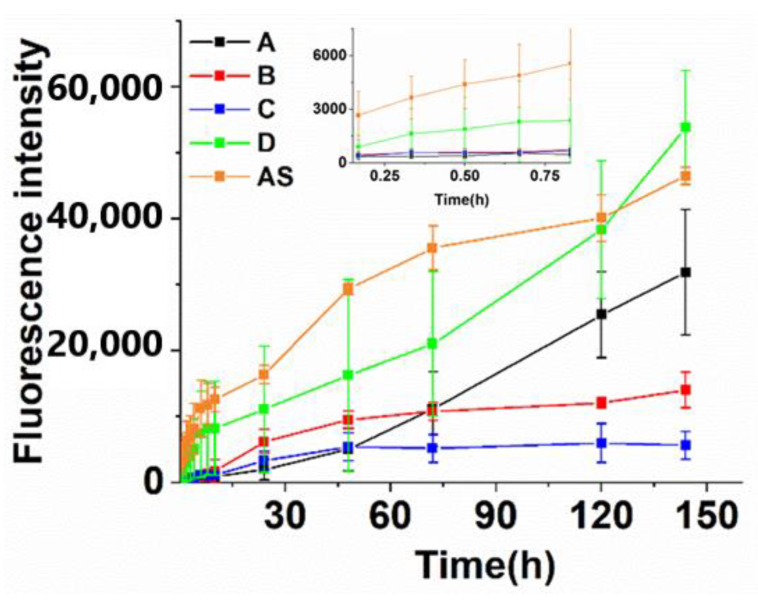
Evaluation of drug (BSA as the representative compound) release profiles in different hydrogels in vitro. (The florescence intensity for 100% release is about 600,000).

**Figure 6 biomolecules-12-01582-f006:**
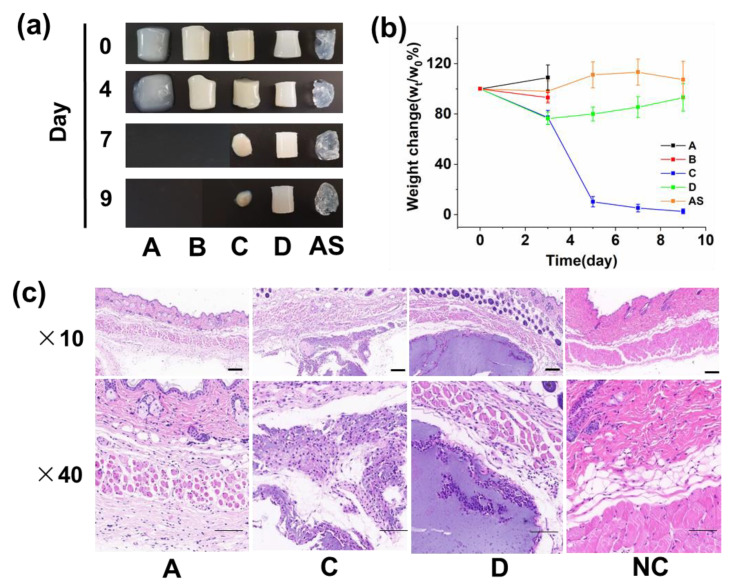
The degradation profile in vitro and in vivo (Scale bar = 200 μm). (**a**,**b**) The hydrogel degradation behaviors in vitro were investigated during a 9-day period. The photographs of the undegraded gel were taken (**a**) and their weight was measured at fixed intervals (**b**,**c**) The biodegradation behaviors in vivo were investigated 20 days after the gels were implanted in the mice.

**Figure 7 biomolecules-12-01582-f007:**
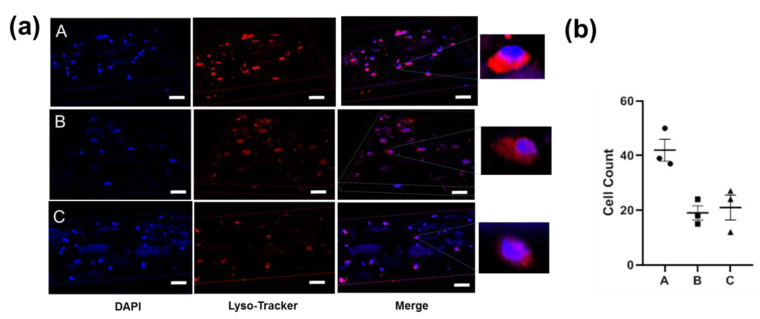
Images of 3D NIH-3T3 cell cultures in different gels in vitro (72 h). (**a**) The cells were stained with DAPI (blue) and lysosome tracker (red). (**b**) The cell count determination in each visual field.

**Figure 8 biomolecules-12-01582-f008:**
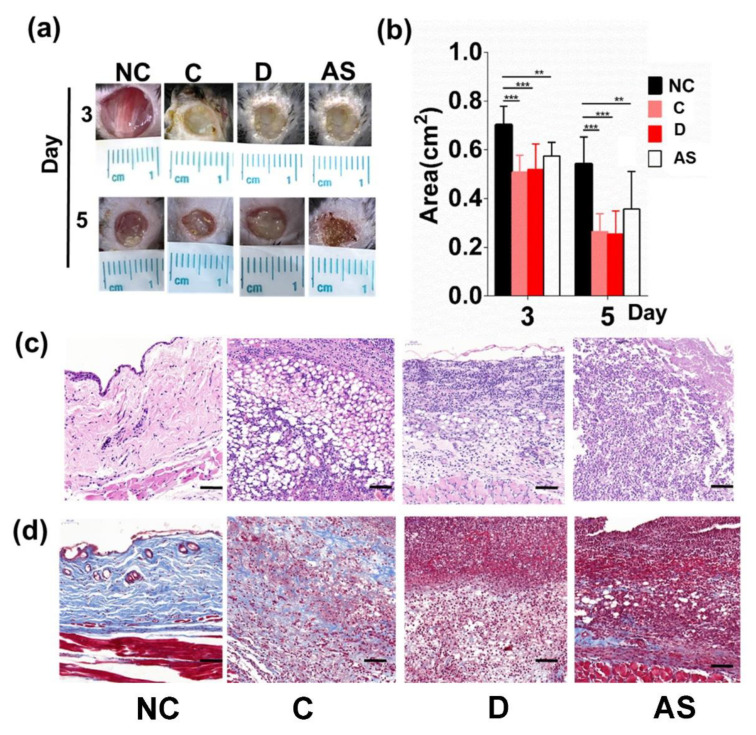
Monitoring the wound-healing process with the hydrogels on a full excision wound of diabetic mice during a 5-day period. (**a**) Digital images of diabetic wounds at Day 3, 5. (**b**) Qualification of the wound area in response to the wound therapies (right panel, *n* = 5) (** *p* < 0.01, *** *p* < 0.001). (**c**) Representative H&E staining of the wound tissues from each group (Day 5). (**d**) Representative staining of the wound tissues with Masson’s trichrome method.

## Data Availability

The data were available on request.

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
