# Peer review of "Precise Design of Alginate Hydrogels Crosslinked with Microgels for Diabetic Wound Healing"

_biomolecules, 2022, doi:10.3390/biom12111582_

Round 1
Reviewer 1 Report
The manuscript describes preparation of alginate hydrogel-based materials using a microgel route to include crosslinks and the subsequent use of the obtained materials for treatment of diabetic wounds. The alginate hydrogels preparation workflow include fabrication of microgels comprised of carboxymethyl chitosan and aldehyde modified hyaluronic acid that also includes zinc. The authors claim that the workflow provides more flexible parameter control to allow tuning of properties like mechanical moduli, degradation and drug release than existing routes. Overall, development of novel wound healing materials offering tailoring properties targeted towards a particular application, are considered of interest for a broad group of readers of the journal. Combined with the overall potential of the presentation, it is believed that the manuscript can be sufficient matured to eventually recommend acceptance for publication. The current reservation is based on issues in the presentation as summarized in the following, that the authors should consider:
11 In the abstract, the authors state “ … tunability of the ironic cross-linking method ..” (line index 12). Is there a typo in this statement? (e.g. should it be ionic instead of ironic, or should it be iron crosslinking?).
22 What is meant by “super high biocompatibility” (line index 29), e.g. as extension of just stating biocompatible?
33 What is meant by “highly dynamic progress..” (line indices 35-36)? Would e.g. “highly dynamic process” be more appropriate?
44 The authors have characterized their samples by various methods thus providing basic information as needed to put the work in context or for others to build upon. Nevertheless, some issues in the presentation of these parameters should by considered: The number of significant figures for data summarized in Table S1 appear to be beyond what the method can provide. Is the last numbers really significant? The units for molar mass should be kDa (kilo prefix abbreviated) and not KDa. The legend of Figure S3 should preferably be explicit on that this is a proton spectrum (similar to as stated in the legend to Figure S2). It is stated in line index 110 that the pH was adjusted also for the sample prepared for the proton NMR characterization. Should this be pD since it is deuterated water?
55 For the characterization of the size of the microgels, it is stated that 100 ng of microgels were dispersed pr mL of water. Is the weight here referring to a dispersion of microgels at a particular concentration?
66 The authors used the wording “electronic interaction” (line indices 136 and 243) when alluding to the nature of driving forces. This is not a widespread wording of the forces involved in such processes, and the authors are encouraged to consider alternatives more in line with the literature.
77 Concerning the determination of the rheological properties of the hydrogels, it appears from the description that the measurements were conducted on hydrogels prepared as such before being loaded in the sample cell. If so, how was the hydrogel specimens adjusted to the actual geometry? If, alternatively, the samples were molded directly in the sample geometry, please describe the process implemented.
88 Also concerning the rheological data: How reproducible were the obtained data? Are the data presented, e.g. Fig. 4f, average over several independent preparations? If not, it is considered essential to check the reproducibility. For the interpretation of the rheological data, the authors state “The results showed the mechanical strength was enhanced as the crosslinking density increased.” What is the crosslinking density in this context? Is the microgel particle concentration alluding to a mechanism where the microgel particle serve as a crosslinker? What is the role of zinc ion introduced as part of the microgel preparation – can that serve as a crosslinker?
99 Concerning the release study data (Fig. 5): What is the scale for 100% release? How many replicates were conducted as basis for the uncertainties in the depicted data? Moreover, related to the same data, the authors allude to a correlation between pore size as determined in SEM and release rate thus implicating possible transport interference between the released compound and the network structures. Why is there such a correlation when the sizes of the pores are rather large as compared to the size of BSA?
Author Response
Responses to the reviewer 1
- The manuscript describes preparation of alginate hydrogel-based materials using a microgel route to include crosslinks and the subsequent use of the obtained materials for treatment of diabetic wounds. The alginate hydrogels preparation workflow include fabrication of microgels comprised of carboxymethyl chitosan and aldehyde modified hyaluronic acid that also includes zinc. The authors claim that the workflow provides more flexible parameter control to allow tuning of properties like mechanical moduli, degradation and drug release than existing routes. Overall, development of novel wound healing materials offering tailoring properties targeted towards a particular application, are considered of interest for a broad group of readers of the journal. Combined with the overall potential of the presentation, it is believed that the manuscript can be sufficient matured to eventually recommend acceptance for publication. The current reservation is based on issues in the presentation as summarized in the following, that the authors should consider:
Answer: Dear reviewer, thank you for your valuable comments to the manuscript. The answers to your questions were listed point by point below. We would appreciate any further suggestions to the manuscript.
- In the abstract, the authors state “ … tunability of the ironic cross-linking method ..” (line index 12). Is there a typo in this statement? (e.g. should it be ionic instead of ironic, or should it be iron crosslinking?).
Answer: The phrase has been changed to iron cross-linking.
3. What is meant by “super high biocompatibility” (line index 29), e.g. as extension of just stating biocompatible?
Answer: The phrase has been changed to high biocompatibility.
4.What is meant by “highly dynamic progress..” (line indices 35-36)? Would e.g. “highly dynamic process” be more appropriate?
Answer: The sentence has been changed to: For one thing, wound healing undergoes exudative, inflammatory, proliferative and regenerative stages sequentially, and a variety of cellular and matrix components were involved in.
- 5. The authors have characterized their samples by various methods thus providing basic information as needed to put the work in context or for others to build upon. Nevertheless, some issues in the presentation of these parameters should by considered: The number of significant figures for data summarized in Table S1 appear to be beyond what the method can provide. Is the last numbers really significant? The units for molar mass should be kDa (kilo prefix abbreviated) and not KDa. The legend of Figure S3 should preferably be explicit on that this is a proton spectrum (similar to as stated in the legend to Figure S2). It is stated in line index 110 that the pH was adjusted also for the sample prepared for the proton NMR characterization. Should this be pD since it is deuterated water?
Answer: Thank you for your suggestions. The last numbers were deleted. KDa has been changed to kDa. The legend of Fig S3 has been changed to “The anomeric proton region of 1H-NMR spectrum for alginate”.
- For the characterization of the size of the microgels, it is stated that 100 ng of microgels were dispersed pr mL of water. Is the weight here referring to a dispersion of microgels at a particular concentration?
Answer: The sentence has been changed to “The microgel solution were diluted with pure water to a concentration of 100 ng/mL.”
- The authors used the wording “electronic interaction” (line indices 136 and 243) when alluding to the nature of driving forces. This is not a widespread wording of the forces involved in such processes, and the authors are encouraged to consider alternatives more in line with the literature.
Answer: The phrase has been changed to “Electrostatic Interactions”.
- Concerning the determination of the rheological properties of the hydrogels, it appears from the description that the measurements were conducted on hydrogels prepared as such before being loaded in the sample cell. If so, how was the hydrogel specimens adjusted to the actual geometry? If, alternatively, the samples were molded directly in the sample geometry, please describe the process implemented.
Answer: Actually, the cell samples were molded directly in the hydrogel geometry because the biomaterials were injectable, and the gel time is less than 10 min. The cells were pre-mixed with the alginate solution at a density of 200, 000 cell/mL. Then different concentration of microgels were added, followed by the addition of GDL solution. The mixture was placed in the incubator, and the cell loaded hydrogel of A, B and C formed within 10 min. The description has been added in the methods of the manuscript.
- Also concerning the rheological data: How reproducible were the obtained data? Are the data presented, e.g. Fig. 4f, average over several independent preparations? If not, it is considered essential to check the reproducibility. For the interpretation of the rheological data, the authors state “The results showed the mechanical strength was enhanced as the crosslinking densityincreased.” What is the crosslinking density in this context? Is the microgel particle concentration alluding to a mechanism where the microgel particle serve as a crosslinker? What is the role of zinc ion introduced as part of the microgel preparation – can that serve as a crosslinker.
Answer: All the data were repeated at least 3 times. The data could be reproduced. The data is the average over several independent preparations. The error bars has been added in the manuscript.
As shown in the reference 15 and 16, the hydrogel were crosslinked through the interaction between the protonated CMCS and alginate. GDL hydrolyzed and released protons, which protonated the amino groups of CMCS. The hydrogels would be formed through the electrostatic interaction between the protonated CMCS and alginate. Besides, the mechanical strength has been enhanced, and the pore size were decreased as the concentration of the microgels were increased. Therefore, we concluded the CMCS containing microgel as a crosslinker. The explanation has been added in the manuscript.
The Zn2+ solution has powerful coordinate ability with Schiff base and carboxylic acid. It was used to precipitate the microgel from the water during microgel preparation. The alginate could form hydrogel with free Zn2+. However, Zn2+ would not work as a crosslinker under this situation, because the alginate and microgel mixture would not form hydrogel without the addition of GDL.
- Concerning the release study data (Fig. 5): What is the scale for 100% release? How many replicates were conducted as basis for the uncertainties in the depicted data? Moreover, related to the same data, the authors allude to a correlation between pore size as determined in SEM and release rate thus implicating possible transport interference between the released compound and the network structures. Why is there such a correlation when the sizes of the pores are rather large as compared to the size of BSA?
Answer: (1) The florescence intensity for 100 % release is about 600000. 3 replicates were conducted for the depicted data.
(2) We correct the possible reason as: more FITC-BSA would be entrapped in the microgels as the concentration increased.
Reviewer 2 Report
The article “Precisely design of alginate hydrogels crosslinked with microgels for diabetic wound healing” has been well written and experiments were opportunely described. it will be suitable for publication after further revisions:
Authors should explain how they determined the higher cell density in A of Figure 7, since according to the figure it seems that the higher cell density it’s showed in C.
To ensure a quantitively estimation of cells number inside the gel authors have to perform an MTT assay to better define cell metabolic activity inside the gel.
Further, the differential lysosome activity in A B C should be discussed.
Line 318. Cells are not pink
Line 327. Cells in dapi cannot be defined “living”
Line 328. the font size must be corrected
Caption of figure 5 is poorly described
Author Response
Response to the reviewer 2
The article “Precisely design of alginate hydrogels crosslinked with microgels for diabetic wound healing” has been well written and experiments were opportunely described. it will be suitable for publication after further revisions:
Answer: Dear reviewer, thank you for your valuable comments to the manuscript. The answers to your questions were listed point by point below. We would appreciate any further suggestions to the manuscript.
Authors should explain how they determined the higher cell density in A of Figure 7, since according to the figure it seems that the higher cell density it’s showed in C.
To ensure a quantitively estimation of cells number inside the gel authors have to perform an MTT assay to better define cell metabolic activity inside the gel.
Answer: The MTT assay was performed to define the cell metabotic activity. However, the gel also react with MTT because of the existence of aldehyde group. The photos have been included in the word file.
Besides, the 3D cell culture pictures have been re-captured, because the free DAPI and lyso-tracer had also been detected with the confocal microscope in the old version of the manuscript. In order to compare the cell density between different groups, we counted the cell one by one in every visual field of 3D cultures.
Further, the differential lysosome activity in A B C should be discussed.
Answer: The representative cells stained with DAPI and Lyso-Tracker has been enlarged. The lysosome activity bad been decreased as the microgel concentration increased, because the percentage of the cells stained with both DAPI and Lyso-Tracker decreased as the microgel concentration increased. The discussion has been added in the manuscript.
Line 318. Cells are not pink
Answer: The word has been changed to DAPI.
Line 327. Cells in dapi cannot be defined “living”
Answer: Living has been deleted.
Line 328. the font size must be corrected
Answer: The font size has been corrected.
Caption of figure 5 is poorly described
Answer: The caption has been changed to Figure 5. Evaluation of drug (BSA as the representative compound) release profiles in different hydrogels in vitro. (The florescence intensity for 100 % release is about 600000.)

Round 2
Reviewer 1 Report
When revising the manuscript, the authors have adequately addressed previously unclear issues, but at the same time, a bit surprised that they mean "iron" crosslinking for the previous unclear statement in the abstract. I certainly agree that crosslinking of alginate with iron ion may have its limitation, but it is considered that ionic crosslinking is more appropriate. The latter wording would also include crosslinking with iron ions, but more generally also include possibility to consider and select different cations for the crosslinking.
The authors are invited to re-consider their statement on "iron" crosslinking in their abstract (as a very minor revision).
Author Response
Dear refree, we certainly accept your suggestions. The phrase has been changed. Thank you again for your valuable comment.